# Longitudinal trajectories of sexual behavior and incident hepatitis C reinfection among men who have sex with men with HIV

Kris Hage[1,2,3]*, Kalongo Hamusonde[4,5], Daniela K. van Santen[1],
Astrid M. Newsum[1,3], Marc van der Valk[3,6], Kees Brinkman[7], Joop E. Arends[8],
Fanny N. Lauw[9], Bart J.A. Rijnders[10], Arne van Eeden[11], Luisa Salazar-Viscaya[4],
Janke Schinkel[12], Anders Boyd[1,2,6], Maria Prins[1,3]

1 Department of Infectious Diseases Research and Prevention, Public Health Service of Amsterdam, Amsterdam, the Netherlands, 2 Amsterdam UMC location University of Amsterdam, Infectious Diseases, Amsterdam, the Netherlands, 3 Department of Infectious diseases, Amsterdam UMC, University of Amsterdam, Amsterdam Institute for Infection and Immunity, Amsterdam, the Netherlands, 4 Department of Infectious Diseases, Inselspital, Bern University Hospital, University of Bern, Bern, Switzerland, 5 Graduate School of Health Sciences, University of Bern, Bern, Switzerland, 6 Stichting hiv monitoring, Amsterdam, the Netherlands, 7 Onze Lieve Vrouwe Gasthuis (OLVG), Department of Internal Medicine, Amsterdam, the Netherlands, 8 Department of Health, Medicine and Life Sciences, Maastricht University, Maastricht, the Netherlands, 9 Medical Centre Jan van Goyen, Department of Internal Medicine, Amsterdam, the Netherlands, 10 Erasmus University Medical Centre, Department of Internal Medicine and Infectious Diseases, Rotterdam, the Netherlands, 11 Department of Internal Medicine, District of Columbia Klinieken Oud Zuid, Amsterdam, the Netherlands, 12 Amsterdam UMC location University of Amsterdam, Department of Medical Microbiology and Infection Prevention, Amsterdam, the Netherlands

☯ These authors contributed equally to this work.
* khage@ggd.amsterdam.nl

## Abstract

Certain key populations have a high risk of hepatitis C virus (HCV) reinfection, which includes men who have sex with men (MSM) with HIV who continue to engage in behaviors associated with HCV acquisition following clearance. Among MSM with HIV, we aimed to identify longitudinal sexual behavior patterns and estimate reinfection risk within identified patterns. MSM with HIV from the longitudinal, prospective Dutch MOSAIC study (2009–2018) at risk for HCV reinfection were included. Follow-up started following HCV clearance. Risk behavior was assessed using the HCV-MOSAIC score (range = 0.0–7.0), where ≥2 indicates high risk of reinfection. Classes were inferred from the mean HCV-MOSAIC score over time using a latent process mixed-effects model with the covariates age, group sex and casual partnership. The association between classes and HCV reinfection risk was assessed using a joint survival model. In total, 123 MSM were included with a median follow-up of 2.7 years [interquartile range (IQR) = 1.2–4.7]. Two classes were identified: one with mostly lower (C1, n = 67) and one with mostly higher risk behavior (C2, n = 56). During follow-up, both classes had considerable variation in HCV-MOSAIC scores (C1, median = 1.1, IQR = 0.0–2.1 and C2, median = 3.0, IQR = 2.0–3.5). HCV reinfection probability was similar between both classes at year 3 of follow-up [C1, 17%, 95% confidence interval (CI) = 11%−35% and C2, 18%, 95%CI = 15%−47%],

---

**Data availability statement:** The data used in this study are not publicly available due to the sensitivity of the information collected, in accordance with ethical and privacy regulations. The data are available upon reasonable request, subject to approval by the principal investigator. Please contact CoordinatieIZOZ@ggd.amsterdam.nl for more information and/or submission requests.

**Funding:** This work was supported by the "Aidsfonds" Netherlands (grant numbers 2008.026, 2013.037), the Netherlands Organization for Health Research and Development (ZonMw) (grant number 522004006) and GGD research funds. The funders had no role in study design, data collection and analysis, decision to publish, or preparation of the manuscript.

**Competing interests:** I have read the journal's policy and the authors of this manuscript have the following competing interests: J. S.'s institution has received research support and consultancy fees from Gilead, and a speakers fee from Janssen Pharmaceuticals, independent from the submitted work. M. P.'s institution has received speakers fees and independent scientific support from Gilead Sciences, Roche, MSD, and Abbvie, outside the submitted work. M. V.'s institution has received consultancy fees from Gilead, MSD and ViiV outside the submitted work and research grants from Gilead, Merck Sharp Dome and ViiV, A.B. has received a speakers fee from Gilead Sciences, independent for the submitted work, grants from MSD and Gilead Sciences, and participated in advisory boards and received travel support from MSD, Janssen-Cilag, Bristol-Myers Squibb, Gilead Sciences, Pfizer, and ViiV Healthcare, outside the submitted work. All other authors report no potential conflicts. All authors have submitted the ICMJE Form for Disclosure of Potential Conflicts of Interest. Conflicts that the editors consider relevant to the content of the manuscript have been disclosed. Our competing interest disclosure does not alter our adherence to PLOS ONE policies on sharing data and materials.

but became higher in C2 than C1 at year 5 (C1, 22%, 95%CI = 13%−39% and C2, 37%, 95%CI = 28%−69%). The variation in risk over time suggests that behavioral assessment is continually needed for early testing, treatment and offering behavioral inventions.

## Introduction

Hepatitis C virus (HCV) is a public health concern affecting approximately 58 million people worldwide in 2020 [1]. In Western European countries, the virus is circulating among men who have sex with men (MSM) with human immunodeficiency virus (HIV) [2,3]. Sexual and drug use behaviors are known to be a common means of HCV transmission in this population, and include more specifically receptive condomless anal sex (CAS), sexual practices that can cause harm to the anorectal mucosa, injecting drug use (IDU) and sharing straws or other equipment for snorting drugs before or during sex [4–9].

Simplified therapy with direct-acting antivirals (DAAs), which provided higher cure rates and fewer side-effects compared to pegylated-interferon based therapy, became widely available in the Netherlands in 2014 [10–12]. Following the high uptake of DAAs, there has been a notable decline in HCV incidence likely owing to the lower numbers of individuals able to transmit the virus [12–16]. However, in the absence of an effective vaccine, studies have demonstrated that reaching HCV elimination among MSM with HIV in this epidemiological setting requires broad access to DAAs, a reduction in behaviors associated with HCV and ongoing HCV surveillance [15,17].

Of concern is the significant risk for HCV reinfection among MSM with HIV [18–20]. While the incidence of primary HCV infection in the Netherlands has seen marked declines following broad access to DAAs [from 1.0/100 person-years (PY) in 2015 to 0.4/100 PYs in 2019], the incidence of HCV reinfection also declined but remained relatively high at 1.1/100 PYs in 2019 [14]. Accordingly, the proportion of HCV incident cases due to reinfection increased from 24% in 2015 to 35% in 2019 [20]. Modelling data have indicated that the majority of HCV infections among MSM with HIV will be reinfections in the future if no decline in behaviors associated with HCV occurs [15]. Prevention measures against HCV reinfection will likely face challenges if some MSM are unwilling or unable to adopt behavioral risk reduction strategies after HCV clearance.

Until recently, studies that have assessed longitudinal behavior following HCV treatment have only focused on a limited number of behaviors associated with HCV acquisition or included different key populations for HCV [e.g., people who inject drugs (PWID) or receive opioid agonist therapy] [21–25]. While there have been studies that linked specific behaviors with HCV reinfection at a single time point, only few examined how these behaviors evolve over time [26]. Groups of MSM who more frequently engage in high-risk behaviors over time following viral clearance may have higher risk of HCV and subsequently onward transmission. MSM with higher levels of behavioral risk may warrant additional testing and could benefit most from behavioral

interventions. Therefore, this study aims to identify classes of MSM with HIV with different patterns of HCV-related behavior over time and estimate the risk of HCV reinfection within these patterns.

## Methods

### Study design and participants

We used data collected in the Dutch MSM Observational Study of Acute Infection with hepatitis C (MOSAIC) study. The MOSAIC study procedures have been described elsewhere [5]. Briefly, the MOSAIC study was a prospective, observational cohort study conducted between 15 May 2009 and 1 December 2018 enrolling MSM with HIV with a confirmed acute HCV infection either prospectively around the moment of HCV diagnosis, or retrospectively at any moment after diagnosis. Socio-demographic, clinical, and virologic data were retrospectively collected from primary HCV infection and after inclusion at each semi-annual study visit together with a detailed self-administered questionnaire containing questions about sexual behavior and substance use in the previous six or twelve months. The MOSAIC study was approved by the Institutional Review Boards of the Academic Medical Center at the University of Amsterdam and ethical committees/board of directors of each institute recruiting participants (NL26485.018.09 and NL48572.018.14). All participants gave written informed consent.

### Study assessments

For the present study, we included participants who were at risk for HCV reinfection (i.e., participants who completed ≥1 questionnaire with behavioral data after HCV clearance). HCV clearance was defined as spontaneous or treatment induced clearance [i.e., sustained virological response (SVR)]. Spontaneous clearance was defined as two consecutive negative HCV-RNA tests among participants who did not receive HCV treatment following a positive HCV-RNA test. The date of spontaneous clearance was estimated as the midpoint between the last positive and first negative HCV-RNA test. SVR was defined as having at least one negative HCV-RNA test 12 weeks after the end of treatment with DAAs or 24 weeks after the end of treatment with pegylated-interferon (Peg-IFN) regimen. HCV reinfection diagnosis was defined as having a positive HCV-RNA test result following HCV clearance. The date of HCV reinfection was estimated as the midpoint between the last negative and the first positive HCV-RNA test. Questions about sexual and drug use behavior referred to the preceding six or twelve months and included the following key questions: (1) if receptive anal sex with sex partner(s) was reported, participants were asked, "Were condoms used during receptive anal sex?", (2) if fisting with sex partner(s) was reported, participants were first asked, "Were gloves used during fisting?", and if the response was "yes", participants were additionally asked, "Were gloves shared during fisting?", (3) if use of sex toys was reported, participants were asked, "Did you use any sex toys that your sexual partner also used?", (4) if snorting drugs was reported, participants were asked, "Were straws shared or did you use a straw that had already been used by someone else?", (5) "Did you or your steady/casual partner bleed somewhere during sex", (6) "Did you rinse your rectum before/after sex with your steady/casual partner?", (7) "Did you inject drugs before or during sex?" and (8) "Which of the following STIs did you have?". We defined sexualized drug use (SDU) as the use of any drug before or during sex. Ulcerative STI was self-reported and defined as a syphilis, herpes, or Lymphogranuloma Venereum infection.

Behavioral risk for HCV reinfection was measured using the HCV-MOSAIC risk score, which was previously developed and validated for primary early HCV infection and HCV reinfection [6,27]. The risk score was calculated for all included participants at each study visit. To calculate the score, the beta coefficients of the six factors mentioned were added when present: receptive CAS (beta = 1.1), sharing of sex toys (beta = 1.2), unprotected fisting (beta = 0.9), IDU (beta = 1.4), sharing of straws when nasally-administered drugs are used (beta = 1.0), and having an ulcerative STI (beta = 1.4) [6,27].

### Statistical analysis

Follow-up began following HCV clearance and continued until date of participant withdrawal, end date of the study (i.e., 28 September 2018) or date of HCV reinfection diagnosis, whichever came first. Sociodemographic, clinical and behavioral

characteristics of participants were described at the first study visit following HCV clearance for those with and without HCV reinfection separately.

To identify classes of MSM with different levels of behavioral risk over time, we employed a latent profile analysis. This method involved a two-step process [28]. First, latent profiles of MSM at risk of HCV reinfection were identified using a latent process mixed-effects model by modelling individual-specific risk scores with the covariates age, engagement in group sex (yes/no) and casual partnership status (yes/no). Second, the risk of HCV reinfection for each identified latent profile was jointly modelled to a class-specific survival model. A baseline risk function using splines with three quantile knots (at 25th, 50th and 75th percentile of time) was employed to estimate the probability of HCV reinfection over the study period for each latent class. To determine the optimal number of latent classes, we estimated latent class models ranging from 1 to 5 and selected the model with lowest Bayesian Information Criterion (BIC) and Akaike Information Criterion (AIC) as the best fitting model [29].

Posterior probabilities of belonging to a specific class were calculated for each individual. Individuals were assigned to the class with the highest posterior probability in further analysis. We examined latent class specific trajectories of the risk score over the study period. We assessed the proportion of persons engaging in individual risk behaviors included in the risk score for each latent class. Demographic, clinical and behavioral characteristics at the first study visit following HCV clearance were described and compared using Pearson $\chi^2$ test, Fisher's Exact or Wilcoxon tests, as appropriate. The incidence rate (IR) per 100 PYs of HCV reinfection and its corresponding 95% confidence interval (CI) were analyzed by dividing the number of incident infections by the amount of person-time, overall and for each class. To test the difference in IR between classes, the incidence rate ratio (IRR) with 95%CI was estimated using Poisson regression, while including a covariate representing classes.

In sensitivity analysis, we based comparisons on the second best-fitting latent class model and compared results to the main analysis. Statistical analyses were performed using R (version 3.6.3, Vienna, Austria) and Stata 17 (Statacorp, College Station, TX, USA).

## Results

### Description of the study population

We included a total of 123 MSM with resolved HCV infection (spontaneously or treatment-induced). One hundred fifteen MSM began follow-up after their first HCV infection, and 8 after their first HCV reinfection. Among those included, 113 (91.8%) MSM had achieved SVR following Peg-IFN treatment (n = 73, 59.3%) or DAAs (n = 40, 32.5%), or spontaneously cleared infection (n = 10, 8.2%). Median age was 45 years [interquartile range (IQR) = 41–50]. The majority of participants was born in the Netherlands (85.4%) and had a college degree or higher (63.4%). Median follow-up time was 2.7 years (IQR = 1.2–4.7). Overall, there were 33 HCV reinfections (IR = 8.7/100 PY, 95%CI = 5.9–12.9) (Table 1).

### Trajectories of HCV-MOSAIC risk score over time

In the latent profile analysis, the 2-class model revealed to be best fitting model (S1 Table). The classes were labelled as following: 'mostly lower risk' (n = 67) and 'mostly higher risk' (n = 56). The median risk score of MSM in the mostly lower risk class was 1.1 (IQR = 0.0–2.1) at first study visit following clearance and 1.1 (IQR = 0.0–1.6) at year 5 of follow-up (Fig 1). For MSM in the mostly higher risk class, median risk score was 2.8 (IQR = 2.0–3.7) at first visit following clearance and remained high at 3.0 (IQR = 2.4–3.5) at year 5 of follow-up. MSM in both classes demonstrated wide variation in risk scores over time. Median follow-up time in years was 2.9 (IQR = 1.3–4.7) and 2.3 (IQR = 1.0–4.6) for the mostly lower and mostly higher risk classes, respectively (p = 0.29) (Table 2). The degree to which homogenous MSM with similar risk scores were grouped together (i.e., entropy) was 0.55 (S1 Table) and the mean posterior probabilities for MSM in the mostly lower and mostly higher risk classes were 0.84 and 0.85, respectively (S2 Table).

**Table 1. Characteristics of MSM with HIV participating in the MOSAIC study at the first study visit at risk for hepatitis C reinfection, 2009-2018, the Netherlands.**

| | Overall (n = 123) | Reinfection status at the end of follow-up | |
| --- | --- | --- | --- |
| | | Not reinfected (n = 90) | Reinfected (n = 33) |
| *Sociodemographic characteristics* | | | |
| Age, years | 45 (41-50) | 46 (41-50) | 43 (39-50) |
| Dutch origin | 105 (85.4) | 76 (84.4) | 29 (87.8) |
| College degree or higher† | 78 (63.4) | 56 (62.2) | 22 (66.7) |
| *HIV and HCV clinical characteristics* | | | |
| CD4 cell count (cells/mm³)‡ | 475 (328-642) | 520 (360-700) | 370 (270-561) |
| Nadir CD4 cell count (cells/mm³)‡ | 436 (290-610) | 460 (340-695) | 339 (250-480) |
| On cART | 115 (93.5) | 85 (94.4) | 30 (90.9) |
| Undetectable HIV RNA (<50 copies/mL) | 108 (87.8) | 82 (91.0) | 26 (78.8) |
| Treatment of cured HCV infection | | | |
| Ribavirin and pegylated interferon | 73 (59.3) | 50 (55.6) | 23 (69.7) |
| DAA +/- ribavirin | 40 (32.5) | 31 (34.4) | 9 (27.3) |
| No treatment (spontaneous clearance) | 10 (8.2) | 9 (10.0) | 1 (3.0) |
| *Sexual behaviors*§ | | | |
| Stable sexual partner | 63 (51.2) | 48 (53.3) | 15 (45.5) |
| Casual partner | 78 (63.4) | 59 (65.6) | 19 (57.6) |
| ≥ 10 casual partners | 47 (38.2) | 36 (40.0) | 11 (33.3) |
| Bleeding during sex with stable partner | 11 (8.9) | 9 (10.0) | 2 (6.1) |
| Bleeding during sex with casual partner | 18 (14.6) | 14 (15.6) | 4 (12.1) |
| SDU | 57 (46.3) | 44 (48.9) | 13 (39.4) |
| Group sex | 59 (48.0) | 47 (52.2) | 12 (36.4) |
| Anal rinsing | 50 (40.7) | 36 (40.0) | 14 (42.4) |
| Follow-up time, years¥ | 2.7 (1.2-4.7) | 2.8 (1.2-4.7) | 2.5 (1.1-3.6) |

Presented are n (%) or median (IQR).

† Missing data; n = 14 among MSM who were not reinfected, n = 2 among MSM who were reinfected

‡ Missing data; n = 3 among MSM who were not reinfected, n = 0 among MSM who were reinfected

§ Sexual behaviors refer to the previous six months.

¥ Until the end of follow-up or until date of HCV reinfection diagnosis for those reinfected.

Abbreviations: cART, combination antiretroviral treatment; DAA, direct acting antivirals; HCV, hepatitis C virus; HIV, human immunodeficiency virus; IQR, interquartile range; MOSAIC, men who have sex with men Observational Study of Acute Infection with hepatitis C; MSM, men who have sex with men; RNA, ribonucleic acid; SDU, sexualized drug use.

Boxes represent the median HCV-MOSAIC risk score and quartiles. The whiskers represent the minimum and maximum scores. The dashed line shows the validated HCV-MOSAIC cut-off ≥2 indicating higher risk for HCV [6]. The black solid lines represent the modelled HCV-MOSAIC risk score over time with splines with 3-knots. The shaded areas represent the corresponding 95% CI. Individuals were assigned to the latent class (k) for which they had the highest posterior probability ($\pi_k$), whereby $\pi_k$ was determined based on maximum likelihood. Abbreviations: CI, confidence interval; HCV, hepatitis C virus; MOSAIC, men who have sex with men Observational Study of Acute Infection with hepatitis C

When examining individual behaviors within classes over time, the yearly proportions of MSM reporting sharing toys (Fig 2A) and unprotected fisting (Fig 2B) were consistently higher in the mostly higher risk class during the 5 years following HCV clearance (median = 28.6%, IQR = 25.6%−31.7% and 37.5%, IQR = 35.0%−44.8%, respectively), compared to the mostly lower risk class (median = 8.5%, IQR = 7.7%−13.2% and 12.8%, IQR = 6.1%−18.1%, respectively). The proportion

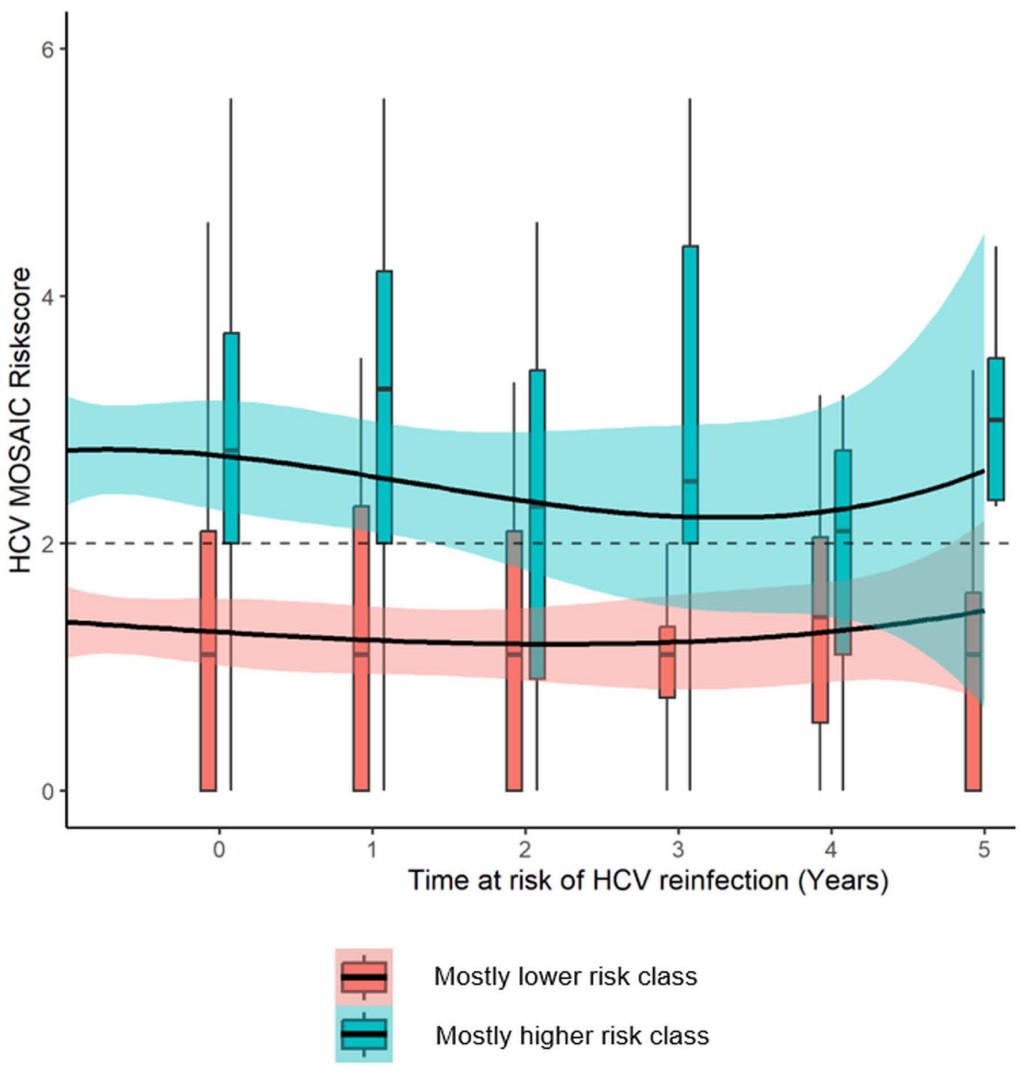

**Fig 1. Trajectories of the HCV-MOSAIC risk score over time per class.**

**Table 2. Characteristics of MSM with HIV participating in the MOSAIC study at the first study visit at risk for hepatitis C reinfection according to membership in the mostly lower and mostly higher risk classes, 2009-2018, the Netherlands.**

| | Mostly lower risk class (n=67) | Mostly higher risk class (n=56) | P |
|---|---|---|---|
| *Sociodemographic characteristics* | | | |
| Age, years | 46 (40-50) | 44 (41-50) | 0.65 |
| Dutch origin | 56 (83.6) | 49 (87.5) | 0.72 |
| College degree or higher† | 37 (55.2) | 34 (60.7) | 0.67 |
| *HIV and HCV clinical characteristics* | | | |
| CD4 cell count (cells/mm$^3$)‡ | 475 (332-675) | 470 (325-628) | 0.91 |
| Nadir CD4 cell count (cells/mm$^3$)‡ | 444 (320-608) | 427 (290-608) | 0.78 |
| On cART | 64 (95.5) | 51 (91.1) | 0.47 |
| Undetectable HIV RNA (<50 copies/mL) | 60 (89.6) | 48 (85.7) | 0.71 |
| Treatment of cured infection | | | 0.63 |
| Ribavirin and pegylated interferon | 41 (61.2) | 32 (57.1) | |
| DAA+/- ribavirin | 22 (32.8) | 18 (32.1) | |
| No treatment (spontaneous clearance) | 4 (6.0) | 6 (10.8) | |
| *Sexual behaviors*§ | | | |
| Stable sexual partner | 31 (46.3) | 32 (57.1) | 0.31 |
| Casual partner | 42 (62.7) | 36 (64.3) | 0.99 |
| ≥10 casual partners | 25 (37.3) | 22 (39.3) | 0.97 |
| Bleeding during sex with stable partner | 2 (3.0) | 9 (16.1) | 0.02 |
| Bleeding during sex with casual partner | 2 (3.0) | 16 (28.6) | <0.01 |
| SDU | 31 (46.3) | 26 (46.4) | 0.99 |
| Group sex | 29 (43.3) | 30 (53.6) | 0.34 |
| Anal rinsing | 22 (32.8) | 28 (50.0) | 0.08 |
| Follow-up time, years¥ | 2.9 (1.3-4.7) | 2.3 (1.0-4.6) | 0.29 |

Presented are *n* (%) or median (IQR).

P-value represents the statistical comparison between reinfection status at first visit at risk for HCV reinfection. Individuals were assigned to the latent class (*k*) for which they had the highest posterior probability (π$_k$), whereby π$_k$ was determined based on maximum likelihood.

† Missing data; n=6 among MSM from the consistent lower risk class, n=10 among MSM from the mostly higher risk class.

‡ Missing data; n=1 among MSM from the consistent lower risk class, n=2 among MSM from the mostly higher risk class.

§ Sexual behaviors refer to the previous six months.

¥ Until the end of follow-up or until date of HCV reinfection diagnosis for those reinfected.

Abbreviations: cART, combination antiretroviral treatment; DAA, direct acting antivirals; HCV, hepatitis C virus; HIV, human immunodeficiency virus; IQR, interquartile range; MOSAIC, men who have sex with men Observational Study of Acute Infection with hepatitis C; MSM, men who have sex with men; RNA, ribonucleic acid; SDU, sexualized drug use.

of MSM reporting sharing of straws in the mostly higher risk class decreased from 34.0% (95%CI=21.5%−46.3%) in the first year after HCV clearance to 12.9% (95%CI=5.8%−20.1%) in year 5 (Fig 2C), whereas in the mostly lower risk class, we observed a decrease from 10.6% (95%CI=1.8%−19.4%) in the first year after HCV clearance to 4.5% (95%CI=0.0%−9.6%) in the second year after HCV clearance. Subsequently, we observed an increase to 15.7% (95%CI=5.7%−25.6%) during the third year after HCV clearance with a decreasing trend afterwards. The proportion of MSM reporting IDU remained low in both classes with a proportion of 7.9% (95%CI=1.2%−4.6%) in the mostly higher risk class and 7.8% (95%CI=0.5%−15.2) in the mostly lower risk class at year 3 after HCV clearance. Thereafter, we observed an increase in the mostly higher risk class to 15.3% (95%CI=7.6%−22.9%) at year 5, whereas in the mostly

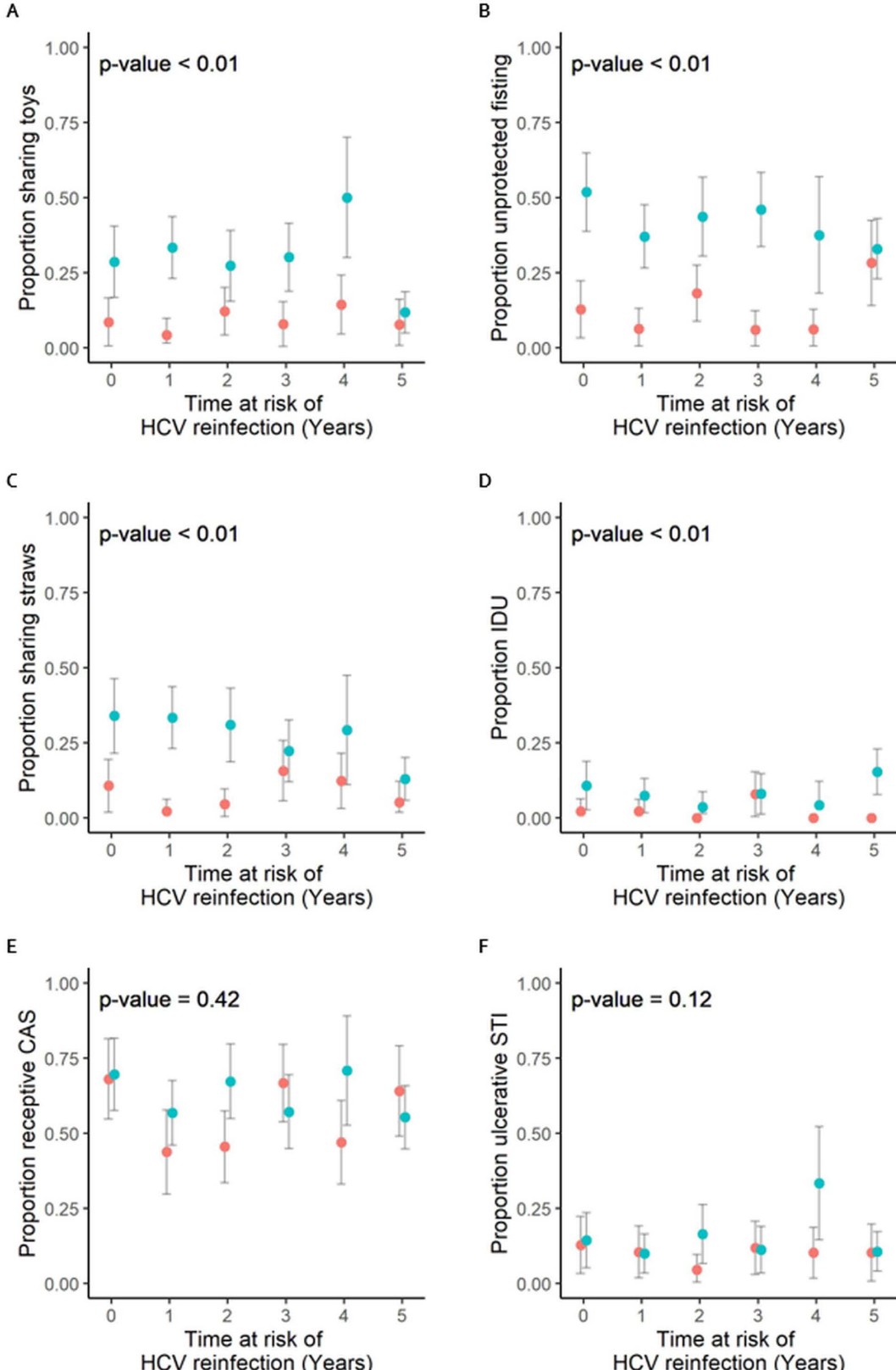

**Fig 2. Proportion of persons reporting individual risk behaviors included in the risk score per assigned class.** P-values indicate overall differences between groups. Bullets and whiskers represent the proportions and corresponding 95% CI of individual risk behaviors for the mostly lower risk

class (orange) and the mostly higher risk class (blue). Individuals were assigned to the latent class (k) for which they had the highest posterior probability (πk), whereby πk was determined based on maximum likelihood. Abbreviations: CAS, condomless anal sex; CI, confidence interval; HCV, hepatitis C virus; IDU, injecting drug use; STI, sexually transmitted infection.

lower risk class the proportion remained low (Fig 2D). The proportion of MSM reporting receptive CAS (Fig 2E) and having an ulcerative STI (Fig 2F) was comparable irrespective of class membership over the study period.

Characteristics of MSM at first study visit following clearance are summarized per class in Table 2. Sociodemographic and clinical characteristics were comparable between classes. The mostly higher risk class reported more bleeding during sex with stable partners compared to the mostly lower risk class (p = 0.02) and more often bleeding during sex with casual partners (p < 0.01). The mostly higher risk class also reported more anal rinsing compared to the mostly lower risk class, albeit this difference was non-significant (p = 0.08).

### Probabilities of HCV reinfection

During follow-up, there were 15 (IR = 6.5/100 PY, 95%CI = 3.6–11.8) and 18 (IR = 11.9/100 PY, 95%CI = 7.0–20.0) HCV reinfections in the mostly lower risk class and the mostly higher risk class, respectively. The rate of having an incident HCV reinfection seemed higher among MSM in the mostly higher risk class compared to MSM in the mostly lower risk class, albeit with wide 95%CI (IRR = 2.0, 95%CI = 0.9–4.5, p = 0.08). The fitted survival probability for remaining without HCV reinfection is depicted in Fig 3. The probability of HCV reinfection was similar at year 3 following viral clearance between the mostly lower risk class (17%, 95%CI = 11%−35%) and the mostly higher risk class (18%, 95%CI = 15%−47%), but became higher in the mostly higher risk class in year 5 compared to the mostly lower risk class (37%, 9%CI = 28%−69% and 22%, 95%CI = 13%−39%, respectively).

### Sensitivity analyses

In sensitivity analyses, the three-class model (with slightly higher entropy) had similar mostly lower (n = 51) and mostly higher risk classes (n = 55), but also included a class of MSM whose mean risk behavior was high in the first two years after clearance and decreased thereafter (n = 17) (S1 Fig). There were, like in the other classes, large variations in risk scores across time points and fewer individuals in this latter class. During follow-up, IR for HCV reinfection were 6.2/100 PYs (95%CI = 3.1–12.4), 12.0/100 PY (95%CI = 7.2–19.8) and 6.3/100 PY (95%CI = 1.6–12.3) in the mostly lower risk, mostly higher risk and initial higher followed by lower risk class, respectively. The IRR for HCV reinfection for the mostly higher risk class and the initial higher class followed by lower risk class compared to the mostly lower risk class was 2.0 (95%CI = 0.9–4.7, p = 0.110) and 1.0 (95%CI = 0.2–4.8, p = 0.986), respectively. The probability of HCV reinfection over time is depicted in S2 Fig.

### Discussion

We studied the association of longitudinal patterns of sexual behaviors linked to HCV reinfection in MSM with HIV. In a context were HCV reinfection was incident, we identified two classes of MSM exhibiting contrasting levels of risk for HCV reinfection. In both classes, MSM demonstrated some variation in risk scores over time, which was mostly the result from changes in sharing sex toys, unprotected fisting, sharing straws when nasally administered drugs were used and receptive CAS. The probability of being HCV reinfected did not differ between classes during the first three years following HCV resolution, but became higher in class exhibiting higher risk behavior as follow-up progressed to year 5.

Variations in risk scores were present in both classes. In the mostly lower risk class, the 75th percentile did fall above the threshold of 2.0 indicating a high risk of HCV reinfection, although their average behavioral risk remained low. Similarly, a study among MSM using PrEP in the Netherlands and Belgium showed varying degrees of sexual behavior over

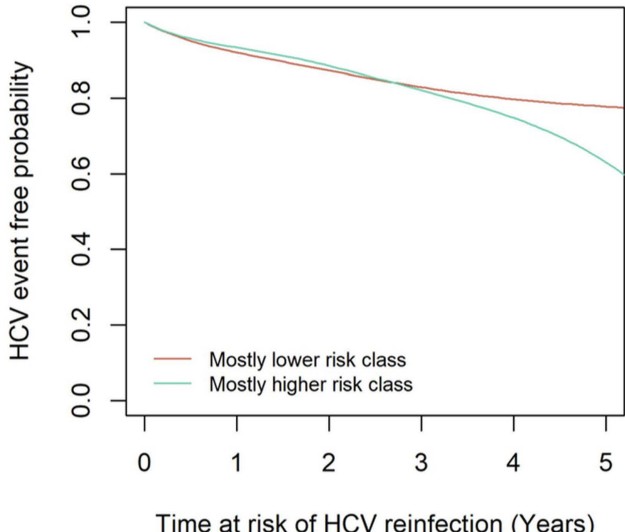

Number of MSM at risk

| | | | | | | |
|---|---|---|---|---|---|---|
| Mostly lower risk class | 67 | 38 | 28 | 20 | 15 | 11 |
| Mostly higher risk class | 56 | 32 | 21 | 13 | 7 | 6 |

**Fig 3. Event-free probability for HCV reinfection over time (in years since at risk for HCV reinfection) per class.** Fitted survival probabilities for remaining without HCV reinfection per assigned class from first visit since viral clearance to year 5 of follow-up. The model was fit using a joint survival model which combines the probability distributions from a linear mixed-effects model with random effects and a survival Cox model. Abbreviations: HCV, hepatitis C virus; MSM, men who have sex with men.

time, where longer periods of activity with lower risk was followed by short periods with high risk [30]. Changes in sexual relationships and being aware that HCV is circulating within an individual's sexual network are important factors related to changes in sexual behaviors [31,32]. Perhaps these factors were related to the variation observed for both classes; however, these data were not collected. In addition, prevention programs with an emphasis on harm and risk reduction were delivered during the DAA era through the NoMoreC project [33]. This intervention could have influenced the progression of the classes hereby identified.

There were no differences in demographic or clinical characteristics between classes at the first visit at risk for HCV reinfection [34]. Part of the lack of differences in demographic characteristics could be due to the rather homogenous study sample. However, although relatively uncommon in our study, we found differences in bleeding during sex with sexual partners between classes, a factor known to possibly facilitate HCV transmission [7]. It is worth noting that prior research has suggested younger age and lower CD4 + cell counts as potential determinants of increased HCV risk, though the latter has yielded mixed results in various studies [35].

There was a two-fold difference in the IR of HCV reinfection between the two classes; however, this difference does have some uncertainty. We showed that HCV reinfection became more frequent in the mostly higher risk class after three

years following viral clearance. There are several possible explanations for this result. Firstly, the median scores between classes overlapped in the first three years of follow-up, making it difficult to fully disentangle risk groups, while MSM in the mostly lower risk class were likely to have periods of high risk. HCV reinfection could have occurred specifically during these periods of high risk. Differences in risk activity became more apparent at later periods of follow-up. In sensitivity analyses with more classes, the variation in risk behaviors was still present and suggests difficulty for the model to separate latent classes when risk might not occur in clear patterns. Secondly, the rates of right-censoring were high after three years of follow-up, and the few participants at later time points could have reduced the statistical power to detect differences between classes. Lastly, there is some evidence that immune protection against HCV reinfection wanes over time following HCV clearance [36]. Perhaps the mostly higher risk class comprised participants with a faster rate of waning immunity, but this hypothesis is highly speculative.

To our knowledge, this is one of the first studies assessing trajectories of HCV-related risk behavior among MSM with HIV who are at-risk for HCV reinfection. Other studies assessing longitudinal behavior following HCV clearance have focused on different key populations or only included receptive CAS and IDU as outcomes, while disregarding the presence of possible distinct trajectories of behaviors within a population [21–24]. For instance, an international study among PWID who were predominantly male and treated with DAAs showed a gradual decline in sharing of injecting equipment although drug use remained stable [22]. A systematic review among PWID from the interferon era reported reductions in IDU following treatment, but there was a high amount of heterogeneity between studies [23]. No reductions in CAS and IDU following treatment were found in a prospective cohort study between 2014–2017 among MSM with HIV/HCV co-infection in Australia [21]. A study that did consider distinct trajectories of behaviors over time among gay and bisexual men and linked these trajectories to incident HCV reinfection showed that those with increasing chemsex probabilities were more likely to acquire an HCV reinfection [26].

Strengths of this study included the prospective study design, which allowed for frequent monitoring of detailed sexual behavior associated with HCV. Nevertheless, there were some limitations. First, we had a relatively limited duration of follow-up among MSM who were at risk for HCV reinfection, although most HCV reinfections tend to occur in the first years after clearance [18]. Second, over half of included MSM were cured with Peg-IFN. In the Netherlands, low barriers for accessing Peg-IFN treatment were in place before DAAs became available in 2014, resulting in a large proportion of individuals with HCV receiving treatment with Peg-IFN during those years. As treatment of HCV is currently fully based on DAAs, our findings may not be directly translatable to the current situation [8]. Third, we found variation in risk behaviors among those who were at higher risk for HCV reinfection. Target groups for behavioral interventions are often selected based on reported risk behavior at a certain time point. Understanding the reasons for changes in sexual risk behavior trajectories would help offer behavioral interventions to MSM before increases in HCV-related risk behavior occur and are retrospectively reported, thereby preventing ongoing HCV transmission. Fourth, while the 2-class model emerged as the best fitting one, the latent class mixture model could have been limited by the variability in sexual behaviors and to a lesser extent, the small sample size. Although we used well established criteria to select the best-fitting model, these criteria could be influenced by model specification or even different functional forms of modeling time [37]. Future studies would be needed to validate these classes, while acknowledging that such data are rarely collected [27]. Fifth, the few numbers of HCV reinfections produced fairly wide CIs for the IRs and survival probabilities, reducing power to effectively assess differences in HCV reinfection rates between classes. Sixth, while we acknowledge the age of the presented data, we believe our findings remain valuable, given that COVID-19 pandemic-related changes in behaviors were transient and that our study population remain comparable to current populations at risk for an HCV infection [38–40]. Furthermore, questions about engaging in sexual and drug use behaviors in the preceding months were self-reported and could therefore be biased by individual recall. Lastly, this study was conducted in a high-income country and predominantly included Dutch MSM with HIV who had at least a college degree. Results might therefore not be generalizable to the wider MSM community with a resolved HCV infection or to settings where the epidemiological context of HCV is different.

In conclusion, we identified two classes of MSM with distinctive patterns of behavioral risk, while variation in risk behaviors seemed to be present in both classes. HCV reinfection risk only diverged between classes after three years following viral clearance. These findings suggest that to inform when HCV testing is most needed, frequent behavioral assessment should be carried out over the years following HCV clearance. In addition, MSM who are suspected of belonging to the trajectory with consistently high levels of HCV-associated behaviors could be offered frequent testing and targeted behavioral interventions to prevent ongoing HCV transmission, as reductions in these behaviors are essential to curb the HCV epidemic in MSM [41]. Further research with longer follow-up times, increased sample size and mixed-method approaches could help clarify how HCV-related behaviors change over time for MSM with HIV and at risk for HCV reinfection.

## Supporting information

**S1 Table. Model fit statistics comparison for latent class analysis.** [†] Model 2 was chosen as the best fitting model based on lowest BIC and AIC. Model description: The best fitting model (2-class model) was based on the lowest BIC and AIC. Entropy is the degree of class separation ranging from 0–1, where an increased value indicates greater ability of the model to categorize persons into clusters. Abbreviations: AIC, Akaike information criterion; BIC, Bayesian information criterion; G, number of classes; NPM, number of estimated parameters.
(DOCX)

**S2 Table. Mean posterior probabilities in each class per LCA model.** [†] Main Analysis. [‡] Sensitivity Analysis. [§] Posterior probabilities are not calculated for a 1-class model. From the latent profile analysis, an *a* posterior probability of an individual *i* belonging to each class is estimated using the maximum likelihood function (i.e., sums of the conditional likelihoods of each latent class, multiplied by the associated latent class probabilities). Mean posterior probabilities are calculated across individuals, indicating the average probability of their membership in the assigned latent class *k*. Higher mean posterior probabilities indicate a higher degree of confidence in the assigned class membership. Abbreviations: LCA, latent class analysis.
(DOCX)

**S1 Fig. Trajectories of the HCV-MOSAIC risk score over time per class.** Boxes represent the median HCV-MOSAIC risk score and quartiles. The whiskers represent the minimum and maximum scores. The dashed line shows the validated HCV-MOSAIC cut-off ≥2 indicating higher risk for HCV [6]. The black solid lines represent the modeled HCV-MOSAIC risk score over time modeling using splines with 3-knots. The shaded areas represent the 95% CI. Individuals were assigned to the latent class (*k*) for which they had the highest posterior probability (πk), whereby πk was determined based on maximum likelihood. Abbreviations: CI, confidence interval; HCV, hepatitis C virus; MOSAIC, men who have sex with men Observational Study of Acute Infection with hepatitis C.
(TIF)

**S2 Fig. Event-free probability for HCV re-infection over time (in years since at risk for HCV re-infection) per class.** Fitted survival probabilities for remaining without HCV reinfection per assigned class from first visit since viral clearance to year 5 of follow-up. The model was fit using a joint survival model which combines the probability distributions from a linear mixed-effects model with random effects and a survival Cox model. Abbreviations: HCV, hepatitis C virus; MSM, men who have sex with men.
(TIF)

## Acknowledgments

The authors thank all participants of the MOSAIC study. In addition, the authors would like to thank Femke Lambers, Joost Vanhommerig, Wendy van der Veldt, Jan van der Meer, Colette Smit, Thijs van de Laar, study nurses, data managers and lab technicians for their contribution to the MOSAIC study.

## Author contributions

**Conceptualization:** Kris Hage, Daniela K. van Santen, Astrid M. Newsum, Marc van der Valk, Anders Boyd, Maria Prins.

**Data curation:** Kees Brinkman, Joop E. Arends, Fanny N. Lauw, Bart J.A. Rijnders, Arne van Eeden, Janke Schinkel.

**Formal analysis:** Kris Hage, Kalongo Hamusonde.

**Methodology:** Kris Hage, Kalongo Hamusonde.

**Supervision:** Anders Boyd, Maria Prins.

**Writing – original draft:** Kris Hage, Kalongo Hamusonde.

**Writing – review & editing:** Daniela K. van Santen, Astrid M. Newsum, Marc van der Valk, Kees Brinkman, Joop E. Arends, Fanny N. Lauw, Bart J.A. Rijnders, Arne van Eeden, Luisa Salazar-Viscaya, Janke Schinkel, Anders Boyd, Maria Prins.

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
