## [Decision Letter · Decision Letter 0]

PONE-D-24-47069Longitudinal trajectories of sexual behavior and incident hepatitis C reinfection among men who have sex with men with HIVPLOS ONE

Dear Dr. Hage,

Thank you for submitting your manuscript to PLOS ONE. After careful consideration, we feel that it has merit but does not fully meet PLOS ONE’s publication criteria as it currently stands. Therefore, we invite you to submit a revised version of the manuscript that addresses the points raised during the review process.

While the findings are intriguing, reviewers have raised concerns regarding the public health implications of these classifications, particularly the lack of targeted intervention strategies. Additional reflection on how these findings can inform or advance public health interventions is necessary. Furthermore, reviewer have raised concern about the data being too old to be relevant at the current time. The authors need to address this issue, and if possible, add recent data to improve study relevance. Finally, reviewers have suggested a number of minor but important revisions to improve clarity, consistency, and precision throughout the manuscript. These suggestions include enhancing the presentation of numerical data, addressing methodological and analytical ambiguities, and elaborating on study limitations.

We look forward to receiving your revised manuscript.

Kind regards,

Syed Hani Abidi

Academic Editor

PLOS ONE

2. Your abstract cannot contain citations. Please only include citations in the body text of the manuscript, and ensure that they remain in ascending numerical order on first mention.

 [This work was supported by the “Aidsfonds” Netherlands (grant numbers 2008.026, 2013.037), the Netherlands Organization for Health Research and Development (ZonMw) (grant number 522004006) and GGD research funds.]. 

[I have read the journal's policy and the authors of this manuscript have the following competing interests: J. S.’s institution has received research support and consultancy fees from Gilead, and a speakers fee from Janssen Pharmaceuticals, independent from the submitted work. M. P.’s institution has received speakers fees and independent scientific support from Gilead Sciences, Roche, MSD, and Abbvie, outside the submitted work. M. V.’s institution has received consultancy fees from Gilead, MSD and ViiV outside the submitted work and research grants from Gilead, Merck Sharp Dome and ViiV, A.B. has received a speakers fee from Gilead Sciences, independent for the submitted work, grants from MSD and Gilead Sciences, and participated in advisory boards and received travel support from MSD, Janssen-Cilag, Bristol-Myers Squibb, Gilead Sciences, Pfizer, and ViiV Healthcare, outside the submitted work. All other authors report no potential conflicts. All authors have submitted the ICMJE Form for Disclosure of Potential Conflicts of Interest. Conflicts that the editors consider relevant to the content of the manuscript have been disclosed.].

5. We note that you have indicated that there are restrictions to data sharing for this study. PLOS only allows data to be available upon request if there are legal or ethical restrictions on sharing data publicly. For more information on unacceptable data access restrictions, please see http://journals.plos.org/plosone/s/data-availability#loc-unacceptable-data-access-restrictions.

6. In the online submission form, you indicated that [Data are available upon reasonable request from the Principle Investigator (M. Prins, mprins@ggd.amsterdam.nl).].

Reviewers' comments:

Reviewer's Responses to Questions

**Comments to the Author**

1. Is the manuscript technically sound, and do the data support the conclusions?

Reviewer #1: No

Reviewer #2: Yes

Reviewer #3: Yes

2. Has the statistical analysis been performed appropriately and rigorously? 

Reviewer #1: Yes

Reviewer #2: Yes

Reviewer #3: Yes

3. Have the authors made all data underlying the findings in their manuscript fully available?

Reviewer #1: Yes

Reviewer #2: Yes

Reviewer #3: Yes

4. Is the manuscript presented in an intelligible fashion and written in standard English?

Reviewer #1: Yes

Reviewer #2: Yes

Reviewer #3: Yes

5. Review Comments to the Author

Reviewer #1: Even though this manuscript is quite interesting, this data is too old to be relevant at the current time. Also, a global pandemic happened, which impacted every aspect of life, including public health risks and interventions.

From as old as 15 years and as early as six years is too long a time.

Reviewer #2: This is an interesting paper that using LCA has identified two classes of MSM with distinctive patterns of behavioral risk whose HCV reinfection risk diverged after 3 years of follow up. The main problem with this paper is what is the advantage for public health interventions of using the classification of MSM versus focusing on risk behaviors, given that the authors do not suggest targeted interventions but "frequent behavioral assessment is needed during care and should be extended over the years following HCV clearance". A further reflections on these implications is necessary for a recommendation for publications.

Reviewer #3: This study presents an analysis of hepatitis C reinfection in a cohort of gay, bisexual, and other men who have sex with men living with HIV in Amsterdam, elucidating risk factors for hepatitis C transmission in this high-risk group. It is well written, innovative in focusing on this target population, and employes rigorous statistical methods to gain a deeper understanding of this phenomenon. The key limitations, including a small sample size and limited generalizability are acknowledged. I have only a few suggestions, as below, and otherwise congratulate the authors on their excellent work.

• Abstract. Here and elsewhere in the manuscript, where there are ranges of proportions presented, it would be more clear for the reader to include a % sign for both numbers in the range, as in “95%CI=11%-35%.” Consider spaces before and after the equal sign, as it seems there is enough room.

• Abstract. In presenting the class 1 and class 2 results, please maintain the same order of the classes throughout, presenting first for C1 and then for C2. It would also be more clear to use consistent punctuation here and elsewhere in the manuscript, either “C1, result” or “C1: result.” Please be sure to add a space after the comma or colon. The numbers are very dense and needlessly run together.

• Introduction, lines 73-74. I would suggest “in 2014” since the verb is became and that happens only once. The alternative would be “have been widely available…since…”

• Introduction, lines 80-83. Important information is given about primary and reinfection rates that is left very vague, with no number presented. Please present numeric estimates of the primary and reinfection rates and the proportion of HCV incident cases from the reference cited.

• Methods, lines 123-124. The question about gloves was apparently not very clear. If participants were asked “Were gloves used during fisting or were gloves shared?” then a yes response could either mean that gloves were used or that gloves were shared. Was there any way to tell if gloves were used but not shared?

• Methods, lines 161-162. “The incidence rate (IR) per 100 person-years (PY) of HCV reinfection and its corresponding 95% confidence interval (CI) was analyzed” – the verb should be “were analyzed” since there are two subjects.

• Discussion, line 291. NAD has not been established as an abbreviation and is not used again. Please spell out.

• Discussion, lines 300-301. Please add “an” before “individual’s sexual network.”

• Discussion, line 303. Please correct “prevention programs…was delivered.”

• Discussion, line 307. Please change “in demographic and clinical characteristics” to in demographic or clinical characteristics.”

• Discussion, lines 311-312. Do the prior studies show younger age and lower CD4 counts to be potential determinants? Please be more specific about the direction of this association seen in other studies.

• Discussion, limitations. It could be more clearly stated that the small sample size and relative low number of events were key limitations. For latent class analysis, 250-300 or more is often mentioned as a minimum sample size.

• Discussion, lines 354-356. Consider adding “in other settings or circumstances” to the end of this sentence.

• Discussion, limitations. Reliance on self-reported behaviors and the potential for bias impacting results is not mentioned.

• Discussion, limitations. Consider referring to PMID: 36398215 and discussing how different approaches to modeling risk over time can have important impact on results in latent class trajectory studies and therefore the results should be considered to be hypothesis-generating and should be confirmed in other studies.

• Table 1. There is a footnote about p values and yet no p values are presented in this table. Please add these, as the comparisons are helpful and p values are included in the other comparison table (Table 2).

6. PLOS authors have the option to publish the peer review history of their article (what does this mean? ). If published, this will include your full peer review and any attached files.

**Do you want your identity to be public for this peer review?** For information about this choice, including consent withdrawal, please see our Privacy Policy .

Reviewer #1: **Yes: ** Arshad Altaf

Reviewer #2: No

Reviewer #3: No

---

## [Author Response · Author response to Decision Letter 1]

7 Apr 2025

Please see attached file "Response to Reviewers" where we have addressed all reviewer and editor comments.

---

## [Decision Letter · Decision Letter 1]

Longitudinal trajectories of sexual behavior and incident hepatitis C reinfection among men who have sex with men with HIV

PONE-D-24-47069R1

Dear Dr. Hage,

We’re pleased to inform you that your manuscript has been judged scientifically suitable for publication and will be formally accepted for publication once it meets all outstanding technical requirements.

Kind regards,

Syed Hani Abidi

Academic Editor

PLOS ONE

Additional Editor Comments (optional):

Reviewers' comments:

Reviewer's Responses to Questions

**Comments to the Author**

1. If the authors have adequately addressed your comments raised in a previous round of review and you feel that this manuscript is now acceptable for publication, you may indicate that here to bypass the “Comments to the Author” section, enter your conflict of interest statement in the “Confidential to Editor” section, and submit your "Accept" recommendation.

Reviewer #2: All comments have been addressed

Reviewer #3: All comments have been addressed

Reviewer #4: All comments have been addressed

2. Is the manuscript technically sound, and do the data support the conclusions?

Reviewer #2: Yes

Reviewer #3: Yes

Reviewer #4: Yes

3. Has the statistical analysis been performed appropriately and rigorously? 

Reviewer #2: Yes

Reviewer #3: Yes

Reviewer #4: Yes

4. Have the authors made all data underlying the findings in their manuscript fully available?

Reviewer #2: Yes

Reviewer #3: Yes

Reviewer #4: Yes

5. Is the manuscript presented in an intelligible fashion and written in standard English?

Reviewer #2: Yes

Reviewer #3: Yes

Reviewer #4: Yes

6. Review Comments to the Author

Reviewer #2: This manuscript has improved after feedback from reviewers as the authors have made changes that clarify and resolve the issues that were suggested. I can recommend this manuscript for publication in this updated version.

Reviewer #3: This is a sound analysis and excellent contribution to the literature on risk factors for hepatitis C among men who have sex with men.

Reviewer #4: Authors have revised the manuscript as per reviewers' comments. One last comment, in the title of the manuscript, please add "of" after the word "incident".

7. PLOS authors have the option to publish the peer review history of their article (what does this mean? ). If published, this will include your full peer review and any attached files.

**Do you want your identity to be public for this peer review?** For information about this choice, including consent withdrawal, please see our Privacy Policy .

Reviewer #2: No

Reviewer #3: **Yes: ** Susan M. Graham

Reviewer #4: No

---

## [Editor Report · Acceptance letter]

PONE-D-24-47069R1

PLOS ONE

Dear Dr. Hage,

I'm pleased to inform you that your manuscript has been deemed suitable for publication in PLOS ONE. Congratulations! Your manuscript is now being handed over to our production team.

Kind regards,

on behalf of

Dr. Syed Hani Abidi

Academic Editor

PLOS ONE